# The Mining of Genetic Loci and the Analysis of Candidate Genes to Identify the Physical and Chemical Markers of Anti-Senescence in Rice

**DOI:** 10.3390/plants12223812

**Published:** 2023-11-09

**Authors:** Wenjing Yin, Zhao Huang, Qianqian Zhong, Luyao Tang, Richeng Wu, Sanfeng Li, Yijian Mao, Xudong Zhu, Changchun Wang, Yuchun Rao, Yuexing Wang

**Affiliations:** 1National Key Laboratory of Rice Biological Breeding, China National Rice Research Institute, Hangzhou 310006, China; ywj0419@zjnu.edu.cn (W.Y.); lsanfeng@126.com (S.L.); yijian891001@163.com (Y.M.); ricezxd@126.com (X.Z.); 2College of Life Sciences, Zhejiang Normal University, Jinhua 321004, China; 18057638109@163.com (Z.H.); zmmz0241223@163.com (Q.Z.); tly16688@163.com (L.T.); wrcbiology@163.com (R.W.)

**Keywords:** rice, superoxide dismutase, malondialdehyde, catalase, anti-senescence, quantitative analysis

## Abstract

Premature senescence is a common occurrence in rice production, and seriously affects rice plants’ nutrient utilization and growth. A total of 120 recombinant inbred lines (RILs) were obtained from successive self-crossing of F_12_ generations derived from Huazhan and Nekken2. The superoxide dismutase (SOD) activity, malondialdehyde (MDA), content and catalase (CAT) activity related to the anti-senescence traits and enzyme activity index of rice were measured for QTL mapping using 4858 SNPs. Thirteen QTLs related to anti-senescence were found, among which the highest LOD score was 5.70. Eighteen anti-senescence-related genes were found in these regions, and ten of them differed significantly between the parents. It was inferred that *LOC_Os01g61500*, *LOC_Os01g61810*, and *LOC_Os04g40130* became involved in the regulation of the anti-senescence molecular network upon upregulation of their expression levels. The identified anti-senescence-related QTLs and candidate genes provide a genetic basis for further research on the mechanism of the molecular network that regulates premature senescence.

## 1. Introduction

Rice (*Oryza sativa* L.) is one of the world’s major food crops, and is critical to agricultural production. Plant growth and development culminate in senescence, which is influenced by multiple environmental and genetic factors [1]. As the most important photosynthetic organ, leaves provide photochemical products and energy-producing substances, which play an essential role in the growth and development of rice [2]. However, abnormal or premature senescence in leaves will seriously affect the nutrient utilization, growth, and development of plants, significantly reducing rice yield and quality. In actual production, the occurrence of premature senescence seriously affects the economic benefits of agricultural production [3]. The initiation of the premature senescence process is mainly characterized by chlorophyll degradation, and the most prominent feature is leaf yellowing [4]. Experimental research shows that delaying and accelerating leaf senescence have different degrees of influence on rice yield and quality; organic matter released from senescent leaves can be recycled, and nutrients can be transported to young leaves, fruits, and seeds during their development [5]. Previous studies have found that when leaves enter the senescence stage, macromolecular substances begin to degrade, and the activity of antioxidant enzymes decreases, which leads to the accumulation of reactive oxygen species (ROSs) and damage to plant cells and tissues [6]. At the same time, the increase in intracellular oxygen free radicals leads to the excessive oxidation of fatty acid chains of membrane lipids, which results in the accumulation of malondialdehyde (MDA) in cells. The excessive accumulation of MDA in cells hinders the activities of proteins, lipids, and chlorophyll, resulting in the loss of many physiological functions of cells, causing cell peroxidation, and accelerating plant senescence. ROSs are important signaling molecules, and a high content of ROSs is a typical characteristic of leaf senescence. However, to prevent the excessive accumulation of ROSs in plants, the redox stress system controls the plant’s production of reactive-oxygen-scavenging enzymes, including superoxide dismutase (SOD) and catalase (CAT) [7]. Therefore, the activities of ROS-scavenging enzymes, such as SOD and CAT, as well as MDA content, can be used as indicators of cell senescence [8,9,10,11]. In addition, some special forms of apoptosis could also improve cell antioxidant levels and senescence resistance [12].

As a special growth and development stage of rice, premature senescence is a quantitative trait controlled by multiple genes, which involves a complex genetic regulatory network, and its mutants are easy to identify and obtain [13]. Researchers have mapped QTLs or functional genes associated with premature senescence in rice using different types of genetic populations or associated mutants.

In recent years, QTLs and genes related to anti-senescence in rice have been gradually discovered and identified, and the functions of anti-senescence genes have been confirmed. Two backcross recombinant inbred line populations were constructed using *indica* IR36 and *japonica* Nekken2 as parents, and a total of six QTLs related to leaf senescence were detected on chromosomes 6 and 9, respectively, taking the chlorophyll content after 25 days of flowering as an indicator of the senescence [13]. Most research shows that stay-green traits have a negative correlation with component and yield traits. Upon assessing the stay-green traits that double the haploid populations of Zhenshan 97 and Wuyujing 2 using six indicators, 46 significant QTLs present in 25 chromosomal regions and 50 gene interactions distributed across 66 loci were detected [14]. Likewise, another study confirmed that stay-green traits were not negatively correlated with yield and grain yield traits. The study identified six QTLs’ degree of greenness by using a recombinant inbred line population. Among them, three QTLs associated with stay-green traits were simultaneously associated with the RM422-RM565 region of chromosome 3 [15]. Functional stay-green retention can delay leaf yellowing and maintain photosynthetic capacity. SNU-SG1 was crossed with two conventional cultivars to determine its inheritance via QTL analysis. Using selective genotyping with F_2_ and recombinant inbred line populations, three major QTLs were detected on chromosomes 7 and 9 in both populations [16].

Many genes related to rice senescence have been cloned, which provides a reference for understanding the molecular mechanism of the rice senescence regulatory network. Premature senescence can limit the growth phase of rice and, thus, its productivity. An *ospse1* mutant showed premature leaf senescence from the booting stage; its premature senescence was controlled by a recessive mutant *ospse1* gene, and this gene may regulate the senescence process of leaves via pectin lyase [17]. It is possible to identify rice leaf senescence-associated genes (SAGs) by exposing rice seedlings to darkness, inducing leaf senescence. Among 14 SAG clones, 11 were found to be associated with both dark-induced and natural leaf senescence. The upregulation of the *Osl2* and *Osh69* genes during senescence could play a role in the process of leaf senescence [18,19,20]. The stay-green mutant *nyc1* localizes the short-chain dehydrogenase/reductase gene *NYC1* with three transmembrane domains on chromosome 1, which plays an important role in regulating light-harvesting complex II and thylakoid membrane degradation during leaf senescence in rice [21]. *ESL1* regulates leaf senescence by participating in purine metabolism induction [22]. The photosynthetic chlorophyll content of *esl2* mutants was found to decreased significantly at the heading and booting stages compared with the wild type, and *ELS2* was mapped between SSR markers RM17122 and swu4-13 on chromosome 4 [23]. Early leaf senescence traits were discovered in the *esl11* mutant, which was mapped to chromosome 7 with a physical range of 143 kb [24]. Researchers predicted and analyzed the interaction of rice leaf senescence proteins using a combination of genomics, proteomics, and bioinformatics, laying an important foundation for cloning and functional research on the regulation of senescence genes [25].

In this study, based on previous fieldwork, a total of 120 recombinant inbred lines (RILs) of Huazhan, Nekken2, and their offspring were used as experimental materials. By measuring and quantifying SOD activity, MDA content, and CAT activity, QTL mapping analysis of quantitative trait loci was conducted to screen out the candidate genes that regulate rice anti-senescence. The expression of these genes was detected in the parents through qRT-PCR, laying a theoretical foundation for elucidating the mechanism of the regulation of premature senescence in rice and cultivating anti-senility rice varieties, to further improve the yield and quality of rice.

Therefore, the QTL mapping and candidate gene analysis of anti-senescence-related traits are helpful methods that can be used to understand this molecular mechanism, further improving rice yield, growth, and development by regulating or delaying senescence, and cultivating anti-senescence and high-yield rice varieties, which is of great significance for meeting the increasing grain demand.

## 2. Results

### 2.1. Phenotype Analysis of Anti-Senescence-Related Traits in Both Parents and RIL Populations

Differences between the two parents were observed using rice leaves to measure the anti-senescence-related index data (SOD activity, MDA content, and CAT activity) after rice ripening. The female parent, Nekken2, had strong anti-senescence ability, with average SOD activity, MDA content, and CAT activity values of 148.1657, 407.4956, and 67.5241, respectively, while the average values of the male parent, Huazhan, were 102.3564, 592.3254, and 42.3825, respectively; the anti-senescence ability of Huazhan appears weaker (Figure 1).

The same method was used in the RIL populations. The results showed that the anti-senescence-related indexes of the 120 RILs presented a continuous normal distribution with a few transgressive individuals, showing the genetic characteristics of quantitative traits and meeting the needs of QTL interval mapping (Appendix A).

### 2.2. Analysis of QTL Mapping Results Related to Anti-Senescence Traits

Based on the constructed high-density SNP molecular linkage map, QTLs for the anti-senescence traits (including SOD, MDA, and CAT) of the RIL populations were mapped. For the above three anti-senescence traits, we detected 13 QTLs with LOD scores above 2.5 (Table 1 and Figure 2).

For SOD activity, Nekken2 provided favorable alleles, with four QTLs detected on chromosomes 3, 5, 8, and 12 in rice, respectively. *qSOD8* showed larger effects, and was located on chromosome 8 between 102.48 and 102.66 cM, with an LOD score of 3.14 and a physical distance of ~43.04 kb. A single QTL was located between 60.87 and 61.18 cM on chromosome 3, with a threshold value of 2.92 and a physical distance of ~71.33 kb. The other two were located on chromosome 5 between 99.15 and 99.44 cM, with a threshold value of 2.50 and a physical distance of ~66.06 kb, and between 91.33 and 92.74 cM, with an LOD score of 2.55 and a physical distance of ~328.7 kb. Since none of the QTL intervals were very large, some of them contributed high LOD scores and may represent the major QTLs of each trait; the others may be minor QTLs, which also affected RIL phenotypic variation.

For MDA content, Nekken2 provided favorable alleles. Seven QTLs were detected and distributed on chromosomes 4, 9, and 11, three of which showed large effects. Among them, the most significant QTL was *qMDA9.2*, which was located between 44.10 and 46.04 cM on chromosome 9 with a threshold as high as 3.58. The other two QTLs, *qMDA4* and *qMDA11*, were found on chromosomes 4 and 11, with LOD scores of 3.56 and 3.33, respectively.

For CAT activity, Nekken2 provided favorable alleles. Two QTLs were detected on chromosomes 1 and 3. Among them, *qCAT1* had a relatively large threshold value and was located between 152.39 and 153.33 cM on chromosome 1, with a threshold value of 5.70. This suggests that *qCAT1* be a major QTL controlling for the CAT trait.

Nekken2 provided favorable alleles for the above three anti-senescence traits, which also confirmed that Nekken2 had stronger anti-senescence ability than Huazhan; this trait could be used in breeding to develop better varieties.

### 2.3. Anti-Senescence Candidate Gene Expression Level Analysis

The candidate genes in the QTL regions for anti-senescence-related traits (SOD activity, MDA content, and CAT activity) in rice were screened, analyzed, and summarized based on the rice genomic database (http://rice.plantbiology.msu.edu/, accessed on 3 July 2023) (Table 2).

We screened all functional genes in 13 QTL intervals on chromosomes 1, 3, 4, 5, 8, 9, 11, and 12 (http://rice.plantbiology.msu.edu/, accessed on 3 July 2023), and selected anti-senescence candidate genes based on their function (http://ricedata.cn/, accessed on 3 July 2023). The first website showed the gene location of all functional genes in order to find all functional genes, and the latter showed all selected gene functions. Then, we selected 18 candidate genes. Proteins encoded by these genes, including the tyrosine protein kinase domain-containing protein, DUF640 domain-containing protein, BAG protein, vacuolar ATP synthase 98 kDA subunit, etc., are shown in Table 2. The expressions of each candidate gene associated with anti-senescence traits in rice were compared via qRT-PCR analysis (Figure 3). Among them, the expression levels of *LOC_Os01g61500*, *LOC_Os01g61810*, *LOC_Os04g40130*, and *LOC_Os09g16950* in Huazhan were significantly higher than those in Nekken2. The expression levels of *LOC_Os08g37760*, *LOC_Os09g16920*, *LOC_Os09g17010*, *LOC_Os11g40690*, *LOC_Os11g40750*, and *LOC_Os12g35330* in Huazhan were significantly lower than those in Nekken2. These anti-senescence candidate genes were highly expressed in parents, indicating that they may participate in antioxidant regulation and affect plant senescence.

## 3. Discussion

Transgressive segregation is a common phenomenon in plant breeding populations, in which hybrids produce offspring that are superior to their parents, and a few recombinants are aberrant in the range of the parental phenotypes, which has been attributed to complementation and epistatic effects [26,27].

In this study, this phenomenon was observed for SOD, MDA, and CAT in the recombinants associated with anti-senescence traits. The extreme phenotypes caused by transboundary segregation will be repaired in the second generation, and the good traits will be stably inherited by the third generation and later undergo recombination, which plays an important role in plant growth and evolution. Three indices related to anti-senescence traits differed significantly between the parents, and they could be used to improve the anti-senescence-related traits of rice via marker-assisted selection, a phenomenon that greatly promote crop improvement from a breeding point of view [28].

### 3.1. QTL Mapping Showed Major and Minor QTLs Related to Anti-Senescence in Rice

The anti-senescence traits of rice are quantitative traits controlled by multiple genes, the expression of which is susceptible to the influence of various factors. The selection of different rice populations, diverse growing environments, and varying treatment methods will all affect the results of QTL mapping. Compared with previous research results, we confirmed the findings some previous studies, and also found some new QTL regions.

QTL regions were detected on chromosomes 1, 3, 4, 5, 8, 9, 11, and 12, with many of them close to or overlapping with previously reported QTLs. For example, *qSOD3* could be related to several stay-green QTLs located on chromosome 3, as reported by Yoo et al. This could be an important region associated with plant senescence, as it improves the chlorophyll content and adjusts the SOD activity [16]. *qSOD8* might be related to a premature senescence mutant, *es-8* [29]. Using a DH population, it was found that *qMDA9.3* might be related to the stay-green trait QTL located between RM257-MRG2533 [14]. *qMDA9.5* overlapped with the elongation height QTL and controlled chlorophyll content [30].

Novel main-effect QTLs were also observed in our research. The *qCAT1* LOD score reached 5.70, suggesting that it is a major site regulating CAT activity. The *qMDA4*, *qMDA9.2*, *qMDA11*, and *qSOD8* LOD scores were all above 3, indicating that these are major anti-senescence-associated sites, and could be used for the improvement of rice varieties. Moreover, there were five MDA content QTLs clustered on chromosome 9, which is consistent with the findings of previous studies and implies there are many clustered MDA content-related genes, but the relationships between them are still unclear. In addition, many genes related to senescence (*LOC_Os09g19510*, *LOC_Os09g19500*, *LOC_Os09g19400*, *LOC_Os09g19390*, *LOC_Os09g19360*) were clustered near *qMDA9.3*. This might be because various location groups and environments were used; alternatively, one individual plant deviation could have caused the QTL region to change slightly, or the LOD threshold of 2.5 may have led to some of the minor QTLs and genes being left out.

### 3.2. Analysis of Anti-Senescence-Related Candidate Genes

*LOC_Os01g61500* (*OsBAG4*) induces programmed cell death and improves resistance to pathogens. The overaccumulation of *OsBAG4* can induce self-activation of the immune response and enhance disease resistance [31]. The E3-BAG module regulates the innate immune response in plants, and balances their defense response and growth. *LOC_Os01g61810* (*OsHAP3A*) is a subunit of the HAP complex, which can combine with the CCAAT sequence in the promoter region to regulate the expression of target genes, regulating the development of chloroplasts [32,33]. *LOC_Os04g40130* encodes a mitochondrial precursor protein that belongs to the *NAC* gene family; it has a variety of functions, including regulating the growth and development and responding to environmental stressors such as heat, cold, salt, drought and aluminum [34,35,36]. It is a cis-acting element of auxin (IAA), gibberellin (GA3), cytokinin (CTK), abscisic acid (ABA), ethylene and kinetin (KT) [37]. *LOC_Os09g16950* encodes the precursor of cysteine-rich receptor kinase (CRK) 25; the CRKs’ functions are connected to development, cell death, immunity and stress response, and they are thought to be involved in redox and ROS control [38]. Studies have shown that the overexpression of *AtCRK4*, *AtCRK5*, *AtCRK19*, and *AtCRK20* leads to cell death resulting from a hypersensitivity reaction (HR) [39,40]. Moreover, mutant *crk5* with constitutive CRK5 expression appeared to accelerate senescence and enhance cell death, which was more obvious when it was exposed to continual darkness and oxidative stress [41]. These genes play essential roles in plant disease resistance, chloroplast development, the stress response and ROS control, and are closely related to plant anti-oxidation and anti-senescence ability. Additionally, the expression of the four abovementioned genes in Huazhan was significantly higher than in Nekken2, and combined CAT and SOD activity in Huazhan was lower than in Nekken2. We speculate that Huazhan expressed these genes in response to oxidative stress.

*LOC_Os08g37760* encodes a C3HC4-type domain zinc finger protein. Zinc finger proteins (ZFPs) perform crucial roles in plant stress tolerance, apoptosis, transcriptional regulation, and protein–protein interactions [42]. Most ZFPs are involved in growth and development, and some are involved in the response to oxidative stress. In Arabidopsis, three C3HC4-type ZFPs have been found, all of which have a function in abiotic stress, and *AtAIRP2* is related to antioxidant regulation. In rice, *EL5* is a C3HC4-type ZFP related to growth and development. *LOC_Os09g16920* (*OsSLD1*) encodes the cytochrome b5 protein, which is an endoplasmic reticulum heme protein that acts as an electron transporter, participates in various redox reactions in cells, and regulates the balance of ROS in response to various stressors. The promoter region of *OsSLD1* includes the abscisic acid response element (ABRE), gibberellin response element (GARE motif), and defense and stress response element (TC-rich-repeat), which may be activated under various stressors, and the expression of *OsSLD1* can be affected by the levels of GA and ABA [43]. *LOC_Os09g17010* encodes lectin receptor-like kinase 1. LecRLK is an important family that plays a key role in stress perception by the lectin receptor and stimulates downstream signaling through the kinase domain, regulating plants’ stress response and development. When plants are exposed to stress, their ROS level are upregulated, and the activity of their antioxidant enzymes, such as APX and CAT, increases [44]. Rice plants with overexpressed *OsLecRLK* showed a higher chlorophyll retention rate under salt stress, which delayed the senescence of leaves and reduced the accumulation of ROS in roots, indicating that lectin receptor-like kinase plays an important role in scavenging reactive oxygen species and in anti-senescence [45]. *LOC_Os11g40690* is cinnamyl alcohol dehydrogenase 4, a key enzyme in the biosynthesis of lignin, an important cell wall component [46]. Meanwhile, the plant’s ability to develop an efficient root system architecture under low-N conditions was further strengthened due to the co-localization of this gene with the QTLs for root number, root length, and chlorate resistance, which are involved in root system architecture [47,48]. *LOC_Os11g40750* putatively expresses a citrate synthase, *LOC_OS12g35330* (*OsGRX28*), which encodes glutathione (GRXs), a key player in maintaining the cellular redox environment. Owing to its ROS scavenging activity, *OsGRX* protects glutamine synthase from oxidative damage and is an integral component in plant growth, chlorophyll catabolism, apoptosis, and senescence, as well as resistance to adversity [49,50]. *OsGRX28* belongs to the CC-type glutenin family, reducing the disulfide bonds of other proteins and maintaining the redox potential of cells. According to a previous study, this gene was previously found to be involved in Mn toxicity tolerance in rice, which depends partly on the antioxidant system under oxidative stress [51]. Moreover, the expression of *OsGRX28* was significantly upregulated after the treatment of 6-Benzylaminopurine (BAP), which also appeared higher in response to abiotic stress treatments like desiccation and salt [52]. According to the qRT-PCR results for these genes, their expression in Nekken2 was higher than in Huazhan, which may be related to their participation in the scavenging of oxygen free radicals and the regulation of senescence in plants. This is consistent with the fact that Nekken2 has higher CAT and SOD activity and lower MDA content than Huazhan.

Quantitative analysis showed that *LOC_Os01g61500*, *LOC_Os01g61810*, and *LOC_Os04g40130* were involved in the regulation of the anti-senescence molecular network in rice as they had higher expression levels than the other candidate genes, thereby affecting the growth and development of rice. Once they have been confirmed through functional studies, these candidate genes can be used to regulate the senescence process in rice. Moreover, with the aid of molecular markers with which they are closely linked, these genes or QTLs can be effectively incorporated to fundamentally solve the internal constraints of rice senescence, contributing to delaying senescence and providing a theoretical basis for breeding anti-premature senescence hybrid rice varieties. This could improve the quality and yield stability of rice, generating economic and social benefits.

## 4. Materials and Methods

### 4.1. Experimental Materials

Huazhan (*Oryza sativa* L. subsp. *indica* cv. ‘Huazhan’) has various advantages, such as stable and high yields and multiple derived varieties, and belongs to varieties of good breeding. Nekken2 (*Oryza sativa* L. subsp. *japonica* cv. ‘Nekken2’) is a conventional *japonica* rice variety that was selected and bred in Japan in the 1990s; it has wide spectrum compatibility and strong spectral affinity, and is widely used in compatibility research and breeding. In this study, the *japonica* cultivar Nekken2 was used as the female parent, and the *indica* rice restorer Huazhan as the male parent. Nekken2 was pollinated by taking Huazhan kernels that grew flowers during rice oxidation at 11 a.m. After a period of growth and development, F_1_ seeds are obtained. The F_1_-generation seeds were planted in a field and self-crossed using the method of single-grain transmission. A total of 120 recombinant inbred lines with stable genotypes and phenotypes were obtained after the 12th generation, which were used for subsequent QTL mapping of the populations [53].

### 4.2. Germination Cultivation and Management of Rice Seeds

A total of 50 seeds from the Huazhan, Nekken2 and each individual plant of the RIL population were selected and shelled, and their surfaces disinfected (70% alcohol for 10 min, repeated once, followed by 10% NaClO for 30 min). Then, they were rinsed several times with deionized water and soaked in water for 2 d, during which the water was changed once to better break the dormancy of the seeds. Then, germination was accelerated for 3 d in a constant-temperature incubator at 37 °C, and the water was changed once per day during the incubation period.

Seeds with basically the same budding were selected to simulate the rice growing environment in an artificial-climate incubator and were hydroponic until the trifoliate stage. The incubator conditions were as follows: a light/dark cycle of 14 h/10 h, temperature of 30 °C, and 75% humidity. After growth to the trifoliate stage, 24 seedlings were selected from both parents and from each line and transplanted into the experimental field, with 6 plant varieties arranged in 4 rows per variety. During this period, we implemented conventional water and fertilizer management, and pest and weed control strategies were carried out.

### 4.3. Analysis of Data on Hormone Treatment and Anti-Senescence-Related Traits in Rice

When rice has grown to the tillering stage, the leaves were subjected to multiple hormonal treatments with 100 mM ethylene precursor (ACC), 100 mM gibberellin (GA3), and 100 mM kinetin(KT). After the first round of spraying, the hormone was sprayed again at time periods of 0.25 h, 0.5 h, 1 h, 2 h, 4 h, 6 h, 8 h, 12 h, and 24 h, respectively.

After rice flag leaves had matured, a chlorophyll meter was used to monitor changes in chlorophyll content every 3 days. When the average chlorophyll content significantly decreased, leaves were taken to measure the content of SOD, MDA, and CAT tissues (the measurement of CAT also requires the determination of tissue total protein concentration). The following kits were purchased from Nanjing Jiancheng Technology Co., Ltd. (Nanjing, China): SOD kit (WST-1 method, Cat No. A001-3-2), MDA kit (TBA method, Cat No. A003-1-2) and CAT kit (colorimetric method, Cat No. A007-1-1). Measurements were performed according to the manufacturers’ instructions (methods for determination of enzyme activity and total protein concentration are shown in Appendix A). Ten plants were monitored for each experiment, and each treatment was repeated at least four times.

### 4.4. Linkage Map Construction and QTL Mapping

Genomic DNA was extracted from young leaves of Huazhan, Nekken2, and 120 RILs using the CTAB (hexadecyltrimethylammonium bromide) method. Barcode multiplex sequencing libraries were constructed as per the manufacturer’s recommendations (Illumina), and paired-end sequencing was performed using an Illumina X-Ten sequencer with a 10× sequencing depth for 120 RILs and a 50× sequencing depth for the RIL parents. Readings were aligned to the Nipponbare version 7 reference genome (http://rice.plantbiology.msu.edu/, accessed on 3 July 2023) using BWA MEM version 0.7.10 [54]. SNP calling and filtration were performed using SAMtools version 1.6 [55]. A circular ideogram displaying complete genome variation information was constructed using Circos version 0.67 [56]. Variation sites amongst the RILs obtained from Huazhan and Nekken2 were compared, and their genotypes determined using a hidden Markov model approach [57]. Consecutive SNP sites within the same genotype were clustered, with those less than 100 kb filtered out. Sequencing data were analyzed to obtain a total of 4858 markers, evenly distributed on 12 chromosomes, to construct the genetic maps [53]. R/QTL was used to locate the QTLs controlling the three physiological measures of anti-senescence [58].

Based on the constructed high-density SNP genetic map with 4858 markers, R-QTL analysis was used to map the QTL intervals, including the physical and chemical indexes of SOD, MDA, and CAT, related to rice anti-senescence traits. QTL analysis was performed using R-QTL [59]. LOD > 2.5 was set as the threshold value to determine the existence of the QTLs. We followed the rules proposed by Mccouch et al. [60] for naming QTLs.

In the localization of the QTLs with rice anti-senescence intervals, the rice genome annotation website (http://rice.uga.edu/cgi-bin/gbrowse/rice/, accessed on 3 July 2023) was used to screen candidate genes related to rice anti-senescence, and information like the function of each gene was further analyzed using the national rice data center web site (http://www.ricedata.cn/gene/, accessed on 3 July 2023) to screen out the candidate genes that may regulate the senescence of rice.

### 4.5. Differential Analysis of Candidate Gene Expression

The leaves of both parents were collected for RNA extraction. Total RNA was extracted from the leaves using RNA isolation kits (Invitrogen, Waltham, MA, USA), and 1 μg of total RNA was aspirated to be reverse-transcribed into cDNA using ReverTra AceTM qPCR RT Master Mix with gDNA Remover (TOYOBO, Osaka, Japan).

Based on the QTL localization, candidate genes associated with anti-senescence in rice were selected. In real-time quantitative PCR (qRT-PCR), SYBR Green Realtime PCR Master Mix (TOYOBO, Osaka, Japan) and primers were used to detect the expression levels of candidate salt-tolerance-related genes, with rice *OsActin* used as the internal reference gene to assess the differential expression of candidate genes related to anti-senescence indicators in rice.

The total volume of the solution used for qRT-PCR was 10 μL, including 2 μL of cDNA template, 1 μL each of upstream and downstream primers (10 μmol/L, Appendix A), 5 μL of SYBR qPCR Mix (TOYOBO, Osaka, Japan) and 1 μL of ddH_2_O.

The qRT-PCR reaction program was as follows: 95 °C for 5 min, followed by 35 cycles of 95 °C for 10 s, 57 °C for 20 s and 72 °C for 20 s. Three replicates were set up for each reaction, and relative quantification was carried out using the 2^−ΔΔCT^ method [61]. The data obtained were analyzed to determine significant differences, with Excel used for *t*-tests and GraphPad Prism 6 for graphical analysis. *p* < 0.05 indicates a significant difference, while *p* < 0.01 indicates a highly significant difference.

## Figures and Tables

**Figure 1 plants-12-03812-f001:**
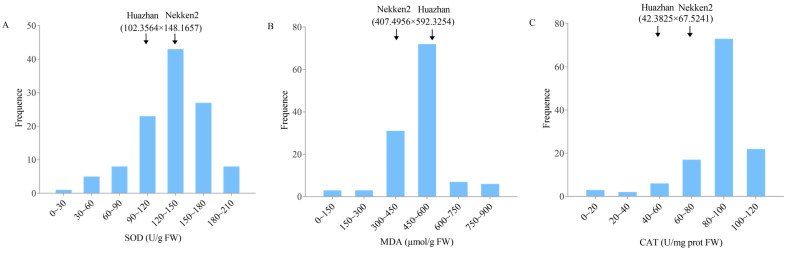
Distribution of anti-senescence related traits in RIL populations. (**A**) SOD activity, (**B**) MDA content, (**C**) CAT activity. FW: fresh weight.

**Figure 2 plants-12-03812-f002:**
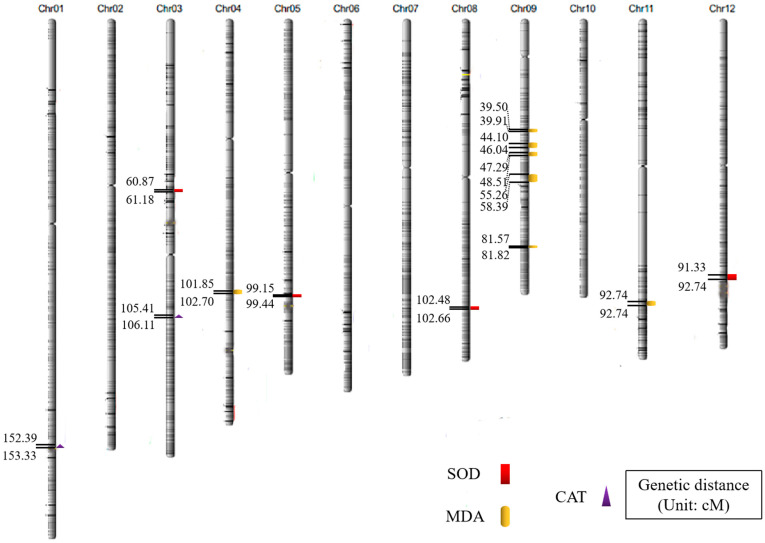
QTL mapping of anti-senescence correlation traits in rice.

**Figure 3 plants-12-03812-f003:**
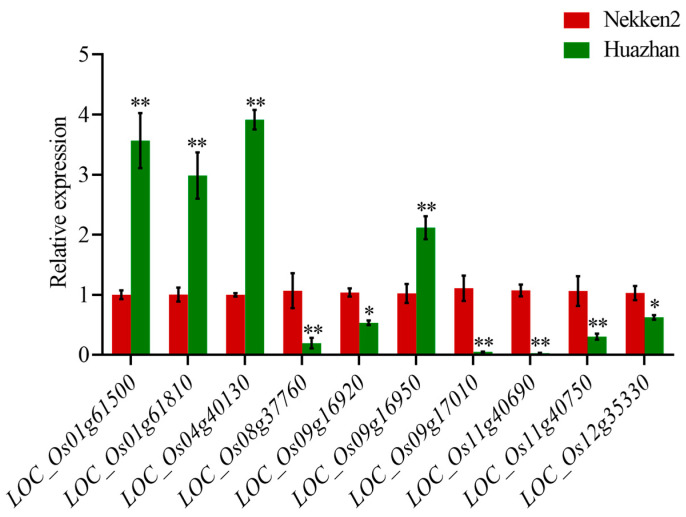
Expression analysis of candidate genes for anti-senescence related traits in rice. * represents a significant difference at the *p* < 0.05 level; ** represents a significant difference at the *p* < 0.01 level.

**Table 1 plants-12-03812-t001:** QTL mapping and effect analysis results of anti-senescence correlation traits in the rice RIL population.

Traits	QTL	Chromosome	Physical Distance (bp)	Position of Support (cM)	LOD
SOD activity	*qSOD3*	3	14,200,134~14,271,458	60.87~61.18	2.92
	*qSOD5*	5	23,130,097~23,196,156	99.15~99.44	2.50
	*qSOD8*	8	23,905,607~23,948,650	102.48~102.66	3.14
	*qSOD12*	12	21,305,692~21,634,437	91.33~92.74	2.55
MDA content	*qMDA4*	4	23,759,183~23,911,317	101.85~102.70	3.56
	*qMDA9.1*	9	9,232,289~9,309,844	39.50~39.91	2.59
	*qMDA9.2*	9	10,288,160~10,740,128	44.10~46.04	3.58
	*qMDA9.3*	9	11,031,216~11,316,376	47.29~48.51	2.86
	*qMDA9.4*	9	12,891,121~13,620,974	55.26~58.39	2.77
	*qMDA9.5*	9	19,029,258~19,086,513	81.57~81.82	2.62
	*qMDA11*	11	24,097,660~24,422,854	103.30~104.69	3.33
CAT activity	*qCAT1*	1	35,548,884~35,768,322	152.39~153.33	5.70
	*qCAT3*	3	24,589,489~24,754,215	105.41~106.11	2.95

**Table 2 plants-12-03812-t002:** Preliminary analysis and summary of function of candidate genes in QTL regions for anti-senescence related traits in rice.

Gene ID	SOD	Gene ID	MDA	Gene ID	CAT
	Putative Function		Putative Function		Putative Function
*LOC_Os03g24930*	tyrosine protein kinase domain containing protein	*LOC_Os04g40130*	Rf1, mitochondrial precursor	*LOC_Os01g61500*	BAG protein
*LOC_Os05g39450*	glutaredoxin	*LOC_Os09g16920*	cytochrome b5	*LOC_Os01g61810*	B subunit of nuclear factor Y
*LOC_Os05g39500*	DUF640 domain containing protein	*LOC_Os09g16950*	cysteine-rich receptor-like protein kinase 25 precursor	*LOC_Os01g61780*	vacuolar ATP synthase 98 kDA subunit
*LOC_Os05g39520*	methyltransferase	*LOC_Os09g17010*	lectin-like receptor kinase 1	*LOC_Os01g61690*	OsSCP5-putative serine carboxypeptidase homologue
*LOC_Os08g37760*	zinc finger, C3HC4 type domain containing protein	*LOC_Os11g40690*	dehydrogenase	*LOC_Os03g43860*	endosomal sorting complex required for transport; ESCRT-III component
*LOC_Os12g35330*	glutaredoxin	*LOC_Os11g40750*	citrate synthase	*LOC_Os03g44000*	LTPL15-Protease inhibitor/seed storage/LTP family protein precursor

## Data Availability

We obtained the QTL mapping and function information of candidate genes from rice gene database (http://rice.plantbiology.msu.edu/, accessed on 3 July 2023), (http://ricedata.cn/, accessed on 3 July 2023), (http://rice.uga.edu/cgi-bin/gbrowse/rice/, accessed on 3 July 2023) and (http://www.ricedata.cn/gene/, accessed on 3 July 2023). All other data supporting this result are included in the article.

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
