# Peer review of "The Mining of Genetic Loci and the Analysis of Candidate Genes to Identify the Physical and Chemical Markers of Anti-Senescence in Rice"

_plants, 2023, doi:10.3390/plants12223812_

Round 1
Reviewer 1 Report (Previous Reviewer 1)
Comments and Suggestions for Authors
The Authors corrected their manuscript according to the reviewers' comments and suggestions. However, a few points should be updated.
line 216 i is "...by Yoo", it has to be removed
line 260 it is "...were lower than Nekken2, we speculated that HuazhanZ expressed those genes to response oxidative stress. " it should be divided on two sentences - were lower than Nekken2. We speculated that HuazhanZ expressed those genes to response oxidative stress.
line 301 there is a lack of the first part of this sentence
Comments on the Quality of English LanguageMinor editing of English language is required.
Author Response
The Authors corrected their manuscript according to the reviewers' comments and suggestions. However, a few points should be updated.
1.line 216 is "...by Yoo", it has to be removed
Response: Thank you for your suggestions. This has been deleted.
2.line 260 it is "...were lower than Nekken2, we speculated that Huazhan expressed those genes to response oxidative stress. " it should be divided on two sentences - were lower than Nekken2. We speculated that Huazhan expressed those genes to response oxidative stress.
Response: Thank you for your suggestions and careful review. According to your suggestion, we have split the sentence here into two sentences.
3.line 301 there is a lack of the first part of this sentence
Response: Thank you for your reminder, we feel really sorry for our careless mistakes. We added the lack of the first part of this sentence.

Reviewer 2 Report (Previous Reviewer 2)
Comments and Suggestions for Authors
The abstract should relate to conclusions. It is still casual.
The Introduction were corrected sufficiently.
Results: the results should relate to tables and figures, but some of them didn't.
The Figure 1: the lack of statistic.
The conclusions should be more precisely (Lines 303-309).
Comments on the Quality of English Language
It's better, but it could be even better.
Author Response
1.The abstract should relate to conclusions. It is still casual.
Response: Thank you for your suggestions and careful review. We re-revised the abstract, make it relevant to the conclusion.
2.The Introduction were corrected sufficiently.
Response: Thank you for your careful review and praise.
3.Results: the results should relate to tables and figures, but some of them didn't.
Response: Thank you very much for your questions. Well, the candidate genes shown in Table 2 were preliminarily screened from 13 QTL intervals, and then we further analysis of the candidate genes in the Table 2, we finally obtained 10 candidate genes in Figure 3, which were quantitatively and functionally analyzed.
4.The Figure 1: the lack of statistic.
Response: Good suggestions, we used the ggqqplot function in the ggpubr package of R language to test the normal distribution of the three groups of data in Figure 1, and found that the distribution of the data was basically near the straight line. Therefore, it can be proved that all of them were in line with the normal distribution. Please refer to Figure S1 in the SUPPLEMENTARY MATERIALS.
5.The conclusions should be more precisely (Lines 303-309).
Response: Thank you for your suggestions and careful review. We revised the conclusion section in more detail.

Reviewer 3 Report (New Reviewer)
Comments and Suggestions for Authors
The work by Yin et al. contributes to the identification of QTL and candidate genes associated to anti senescence in rice. Although the work makes an interesting contribution in the identification of regions associated to leaf senescence reported in previous works as in the identification of novel QTL and candidate genes associated to the senescence process, there are still in this version some concerns to be attended.
1.Although there is an improvement of the English style in the present version, grammar should be carefully revised along the whole manuscript. Some suggestions for corrections are included below but the whole text requires a serious revision.
Introduction:
The experimental research shows that delaying or accelerating leaf senescence of rice yield and quality have different degrees of influence, during leaf senescence, the release of organic matter in the leaves from the senescence can recycle and transport nutrients to the young leafs growth, fruit and seed development [5]. Correction Suggested: “The experimental research shows that delaying or accelerating leaf senescence have different degrees of influence in rice yield and quality; the release of organic matter from senescence leaves can be recycle and nutrients can be transport to the young leaves, fruits and seeds in development [5]”.
As a special growth and development stage of rice, premature senescence is a quantitative trait controlled by multiple genes, which is controlled by a complex genetic regulatory network, and its mutants are easy to identify and obtain [13]. Correction Suggested: “As a special growth and development stage of rice, premature senescence is a quantitative trait controlled by multiple genes, involving a complex genetic regulatory network, and its mutants are easy to identify and obtain [13].”
Two backcross recombinant inbred lines populations were constructed using indica IR36 and japonica Nekken2 as parents, and a total of six QTLs related to leaf senescence were detected on chromosomes 6 and 9, respectively, using chlorophyll content after 25 days of flowering as an indicator of senescence [13]. Correction Suggested: include the word “populations” as indicated above.
Assessing the stay-green traits which double haploid populations of Zhenshan 97 and Wuyujing 2 using 6 indicators, and found 46 significant QTLs present in 25 chromosomal regions and 50 genes interactions distributed across 66 loci were detected [14]. Likewise, another study confirmed that stay-green traits were not or negatively correlated to yield and grain yield traits. Correction Suggested: “Assessing the stay-green traits with double haploid populations of Zhenshan 97 and Wuyujing 2, using 6 indicators, 46 significant QTLs present in 25 chromosomal regions and 50 genes interactions distributed across 66 loci, were detected [14]. Likewise, another study confirmed that stay-green traits were not negatively correlated to yield and grain yield traits”.
Functional stay-green retention can delay leaf yellow and maintain photosynthetic capacity. SNU- SG1 was crossed with two conventional cultivars to determine the inheritance mode. For QTL analysis, using selective genotyping with F2 and recombinant inbred line populations, and 3 major QTLs were detected on chromosomes 7 and 9 in both populations [16]. Correction Suggested: “Functional stay-green retention can delay leaf yellow and maintain photosynthetic capacity. SNU- SG1 was crossed with two conventional cultivars to determine the inheritance mode by QTL analysis. Using selective genotyping with F2 and recombinant inbred line populations, 3 major QTLs on chromosomes 7 and 9 were detected in both populations [16].”
Discussion:
QTL regions located on chromosome 1, 3, 4, 5, 8, 9, 11 and 12. And many of them were close to or overlapped with the QTLs previously studied. For example, qSOD3 could related to several stay-green QTLs located on chromosome 3 by Yoo. It’s implied there was an important region to affect plant senescence by improving chlorophyll content and adjusting SOD activity [16]. Correction Suggested: “QTL regions located on chromosome 1, 3, 4, 5, 8, 9, 11 and 12 were detected, being many of them close to or overlapped with the QTLs previously reported. For example, qSOD3 could be related to several stay-green QTLs located on chromosome 3, reported by Yoo et al. This could be an important region associated to plant senescence by improving chlorophyll content and adjusting SOD activity [16].”
Once confirmed, these genes can be used to regulate the senescence process of rice, and with the aid of molecular markers closely linked with them, these genes or QTLs can be effectively aggregated, so as to fundamentally solve the internal constraints of rice senescence, effectively delay senescence and provide a theoretical basis for breeding anti-premature senescence hybrid rice varieties, thereby realizing the high-quality, high and stable yield of rice and generating economic and social benefits. Correction Suggested: “Once confirmed by functional studies , these candidate genes can be further used to regulate the senescence process of rice, and with the aid of molecular markers closely linked with them, these genes or QTLs can be effectively incorporated, so as to fundamentally solve the internal constraints of rice senescence, contributing to delay senescence and provide a theoretical basis for breeding anti-premature senescence hybrid rice varieties, thereby impacting in high-quality, high and stable yield of rice and generating economic and social benefits.”
2. There are some concerns related to Materials and methods and Results sections that need to be clarified and/or further explained.
Materials and Methods:
Experimental materials:
Field experiment design should be explained with more detail.
Number of plants sampled from each RIL family as well as the .number of biological and technical samples for each experimental measurement should be include in detailed.
Differential analysis of candidate gene expression:
Referred to previous validation of the reference gene used for expression analysis by qRT-PCR
Results
Anti-senescence candidate genes expression level analysis
As mentioned above, it is important to referred to previous validation of the reference gene used in expression analysis under the experimental conditions and developmental phase in evaluation. Include a justification of using only one candidate gene taking into consideration that reference gene expression can varied its expression under different experimental condition and thus differential gene expression results can be less robust compare to the use of more than one reference gene in the analyses.
Comments on the Quality of English LanguageComments on the English Language are included above
Author Response
The work by Yin et al. contributes to the identification of QTL and candidate genes associated to anti senescence in rice. Although the work makes an interesting contribution in the identification of regions associated to leaf senescence reported in previous works as in the identification of novel QTL and candidate genes associated to the senescence process, there are still in this version some concerns to be attended.
Although there is an improvement of the English style in the present version, grammar should be carefully revised along the whole manuscript. Some suggestions for corrections are included below but the whole text requires a serious revision.
Introduction:
1.The experimental research shows that delaying or accelerating leaf senescence of rice yield and quality have different degrees of influence, during leaf senescence, the release of organic matter in the leaves from the senescence can recycle and transport nutrients to the young leafs growth, fruit and seed development [5]. Correction Suggested: “The experimental research shows that delaying or accelerating leaf senescence have different degrees of influence in rice yield and quality; the release of organic matter from senescence leaves can be recycle and nutrients can be transport to the young leaves, fruits and seeds in development [5]”.
Response: Thank you for your professional opinion. We apologize for our inappropriate description. We have made the changes as requested.
2.As a special growth and development stage of rice, premature senescence is a quantitative trait controlled by multiple genes, which is controlled by a complex genetic regulatory network, and its mutants are easy to identify and obtain [13]. Correction Suggested: “As a special growth and development stage of rice, premature senescence is a quantitative trait controlled by multiple genes, involving a complex genetic regulatory network, and its mutants are easy to identify and obtain [13].”
Response: Good suggestion, I am sorry for our inaccurate statement, has been revised.
3.Two backcross recombinant inbred lines populations were constructed using indica IR36 and japonica Nekken2 as parents, and a total of six QTLs related to leaf senescence were detected on chromosomes 6 and 9, respectively, using chlorophyll content after 25 days of flowering as an indicator of senescence [13]. Correction Suggested: include the word “populations” as indicated above.
Response: Thank you for your professional opinion. We apologize for our inappropriate description. We have made the changes as requested.
4.Assessing the stay-green traits which double haploid populations of Zhenshan 97 and Wuyujing 2 using 6 indicators, and found 46 significant QTLs present in 25 chromosomal regions and 50 genes interactions distributed across 66 loci were detected [14]. Likewise, another study confirmed that stay-green traits were not or negatively correlated to yield and grain yield traits. Correction Suggested: “Assessing the stay-green traits with double haploid populations of Zhenshan 97 and Wuyujing 2, using 6 indicators, 46 significant QTLs present in 25 chromosomal regions and 50 genes interactions distributed across 66 loci, were detected [14]. Likewise, another study confirmed that stay-green traits were not negatively correlated to yield and grain yield traits”.
Response: Thank you for your suggestions. We have made the changes as requested.
5.Functional stay-green retention can delay leaf yellow and maintain photosynthetic capacity. SNU- SG1 was crossed with two conventional cultivars to determine the inheritance mode. For QTL analysis, using selective genotyping with F2 and recombinant inbred line populations, and 3 major QTLs were detected on chromosomes 7 and 9 in both populations [16]. Correction Suggested: “Functional stay-green retention can delay leaf yellow and maintain photosynthetic capacity. SNU- SG1 was crossed with two conventional cultivars to determine the inheritance mode by QTL analysis. Using selective genotyping with F2 and recombinant inbred line populations, 3 major QTLs on chromosomes 7 and 9 were detected in both populations [16].”
Response: Good suggestion, I am sorry for our inaccurate statement, has been revised.
Discussion:
6.QTL regions located on chromosome 1, 3, 4, 5, 8, 9, 11 and 12. And many of them were close to or overlapped with the QTLs previously studied. For example, qSOD3 could related to several stay-green QTLs located on chromosome 3 by Yoo. It’s implied there was an important region to affect plant senescence by improving chlorophyll content and adjusting SOD activity [16]. Correction Suggested: “QTL regions located on chromosome 1, 3, 4, 5, 8, 9, 11 and 12 were detected, being many of them close to or overlapped with the QTLs previously reported. For example, qSOD3 could be related to several stay-green QTLs located on chromosome 3, reported by Yoo et al. This could be an important region associated to plant senescence by improving chlorophyll content and adjusting SOD activity [16].”
Response: Thank you for your professional opinion. We apologize for our inappropriate description.. We have made the changes as requested.
7.Once confirmed, these genes can be used to regulate the senescence process of rice, and with the aid of molecular markers closely linked with them, these genes or QTLs can be effectively aggregated, so as to fundamentally solve the internal constraints of rice senescence, effectively delay senescence and provide a theoretical basis for breeding anti-premature senescence hybrid rice varieties, thereby realizing the high-quality, high and stable yield of rice and generating economic and social benefits. Correction Suggested: “Once confirmed by functional studies , these candidate genes can be further used to regulate the senescence process of rice, and with the aid of molecular markers closely linked with them, these genes or QTLs can be effectively incorporated, so as to fundamentally solve the internal constraints of rice senescence, contributing to delay senescence and provide a theoretical basis for breeding anti-premature senescence hybrid rice varieties, thereby impacting in high-quality, high and stable yield of rice and generating economic and social benefits.”
Response: Thank you for your suggestions. We have made the changes as requested. Thank you again.
There are some concerns related to Materials and methods and Results sections that need to be clarified and/or further explained.
Materials and Methods:
Experimental materials:
8.Field experiment design should be explained with more detail.
Response: Thank you for your suggestions and careful review. We made modifications to re-detail the field trial design to make it more easier for readers to understand and read.
9.Number of plants sampled from each RIL family as well as the .number of biological and technical samples for each experimental measurement should be include in detailed.
Response: Thank you for your suggestions. We added the number of plants sampled from each RIL family as well as the number of biological and technical samples for each experimental measurement in 4.3 Analysis of data on hormone treatment and anti-senscence related traits in rice.
Differential analysis of candidate gene expression:
Referred to previous validation of the reference gene used for expression analysis by qRT-PCR
Results
10.Anti-senescence candidate genes expression level analysis
As mentioned above, it is important to referred to previous validation of the reference gene used in expression analysis under the experimental conditions and developmental phase in evaluation. Include a justification of using only one candidate gene taking into consideration that reference gene expression can varied its expression under different experimental condition and thus differential gene expression results can be less robust compare to the use of more than one reference gene in the analyses.
Response: Thank you for your suggestions. The reference genes selected for this study were identified after previously tested and screened in the laboratory. This reference gene was also used for fluorescence quantitative detection of QTL for cadmium accumulation (Pan et al. QTL mapping of candidate genes involved in Cd accumulation in rice grain. 2021), leaf morphology (Wang et al. QTL mapping and analysis of candidate genes in flag leaf morphology in rice. 2021.), watering-tolerance (Rao et al. Identifying of QTL forresistance to submergence in rice. 2020.), heading period (Wei et al. QTL mapping of candidate genes for heading date in rice. 2022.), resistance to stripe disease (Ma et al. QTL exploration of bacterial leaf streak and their gene expression in rice. 2018.), bacterial Diseases (Fang et al. Identification of QTLs conferring resistance to bacterial diseases in rice. 2023) and brown rice rate et al. (Lu et al. Identification of QTL brown rice rate to submergence in rice. 2022.) in rice. All the above studies indicate that this reference gene is stable under biotic or abiotic stresses and is not affected by internal or external factors at different stages. Therefore, this reference gene was used as the standard in this study.

Reviewer 4 Report (New Reviewer)
Comments and Suggestions for Authors
This study identified QTLs and candidate genes for anti-senescence marker phenotypes. A total of 13 QTLs were detected from the F12 RIL population, and candidate gene analysis was conducted via RNA expression analysis. The methodology is clear, and the results and discussion are well organized.
However, the uploaded PDF file includes a manuscript editing track function, making it challenging to read the introduction part. I kindly request the author to re-upload the manuscript.
Below are the major revisions:
1.In Figure 1, it is recommended to show the standard deviation or error of the parental lines on the histogram. Alternatively, provide basic statistics for the F12 population such as mean, standard deviation, and data range.
2.An explanation about the genetic map is necessary. Please include the total genetic distance and average distance between markers. I noticed a significant gap between markers on chromosome 11. It would be helpful if the authors addressed this issue in the linkage map description.
3.In Table 1, it is important to provide the additive effect for each QTL to understand their impact and identify which parent contributes to increasing the value.
4.The section '2.2. Analysis of QTL mapping results related to anti-senescence traits' appears somewhat brief. I suggest adding more explanation. Additionally, qMDA9.19.4 were mapped within a 34 Mb region. Do you consider this region a clustered QTL region or a single locus?
5.Authors should clarify how many genes are located within the QTL region and provide a logical explanation for the selection of these candidate genes. Given the possibility of numerous genes within the QTL region, only 10 genes may not be representative enough to explain the phenotypic variation.
6.The histogram shows transgressive segregation. A discussion is needed to explain this distribution.
Author Response
This study identified QTLs and candidate genes for anti-senescence marker phenotypes. A total of 13 QTLs were detected from the F12 RIL population, and candidate gene analysis was conducted via RNA expression analysis. The methodology is clear, and the results and discussion are well organized.
However, the uploaded PDF file includes a manuscript editing track function, making it challenging to read the introduction part. I kindly request the author to re-upload the manuscript.
Below are the major revisions:
1.In Figure 1, it is recommended to show the standard deviation or error of the parental lines on the histogram. Alternatively, provide basic statistics for the F12 population such as mean, standard deviation, and data range.
Response: Thank you for your suggestions. In the histogram of Figure 1, the ordinate represents the number of occurrences in 120 recombinant inbred lines, that is, the frequency, and the abscissa represents the anti-senescence enzyme activity. The number of occurrences is an exact numerical value, so there is no mean, standard deviation, and data range.
2.An explanation about the genetic map is necessary. Please include the total genetic distance and average distance between markers. I noticed a significant gap between markers on chromosome 11. It would be helpful if the authors addressed this issue in the linkage map description.
Response: Thank you for your suggestions, this is a good questions. In this study, the recombinant inbred line genetic population and related SNP molecular marker genetic map constructed with Huazhan and Nekken2 as parents are the existing basis of this experiment, and these markers are located at a distance of 1.4 cM evenly distributed across 12 chromosomes. Relevant articles on the construction method have been published, for reference: Wang, Y. X., Shang, L. G., Yu, H., Zeng, L. J., Hu, J., Ni, S. et al. (2020). A strigolactone biosynthesis gene contributed to the green revolution in rice. Mol Plant. 13(6), 923-932. doi: 10.1016/j.molp.2020.03.009. This literature is also cited in my manuscript.
3.In Table 1, it is important to provide the additive effect for each QTL to understand their impact and identify which parent contributes to increasing the value.
Response: Thank you for your suggestions, since we used R/QTL method to locate the QTLs, This method didn’t contain R squre values and additive effect values, and LOD score was also an credible index to estimate whether the QTL was a major locus controlling quantitative traits (Arends et al., 2010; Ren et al., 2016). In our research, we detected several QTLs which LOD score were above 2.5, indicated they were credible major effect site.
4.The section '2.2. Analysis of QTL mapping results related to anti-senescence traits' appears somewhat brief. I suggest adding more explanation. Additionally, qMDA9.1, 9.4 were mapped within a 34 Mb region. Do you consider this region a clustered QTL region or a single locus?
Response: Thank you for your suggestions, this is a good questions. We reanalyzed the section '2.2 Analysis of QTL mapping results related to anti-senescence traits' and elaborated in more detail to make it easier for readers. Both of these scenarios are possible. The QTL loci and functional genes related to anti-senescence traits are distributed on all chromosomes, so we prefer this region is a single locus. However, multiple QTL loci were found on chromosome 9 in this study, indicating that there are most likely major QTL loci and candidate genes on this chromosome.
5.Authors should clarify how many genes are located within the QTL region and provide a logical explanation for the selection of these candidate genes. Given the possibility of numerous genes within the QTL region, only 10 genes may not be representative enough to explain the phenotypic variation.
Response: Thank you for your suggestions and careful review. A total of 136 functional genes were identified in the 13 intervals mapped in this study. We screened the candidate genes in the interval by literature review, gene function prediction and annotation, and a total of 18 candidate genes related to anti-senescence were obtained. Finally, through qRT-PCR detection, 10 candidate genes with the most obvious difference in expression between Huazhan and Nekken2 were identified as the final candidate genes for subsequent analysis.
6.The histogram shows transgressive segregation. A discussion is needed to explain this distribution.
Response: Good suggestion, we discuss that why the transgression segregation occurred in the Huazhan and Nekken2 heterozygous progeny in Discussion.

Round 2
Reviewer 2 Report (Previous Reviewer 2)
Comments and Suggestions for Authors
The work has been corrected sufficiently.
Comments on the Quality of English LanguageI hope the minor editing should correct the fluence of language.
Reviewer 4 Report (New Reviewer)
Comments and Suggestions for Authors
Thank you for the nice response for the comments.
This manuscript is a resubmission of an earlier submission. The following is a list of the peer review reports and author responses from that submission.
Round 1
Reviewer 1 Report
Comments and Suggestions for Authors
Please find the attachment with some details of the necessary corrections.
The manuscript "Mining of genetic loci and analysis of candidate genes for physical and chemical markers of anti-senescence in rice" has the potential to be an interesting paper for Plants readers, but it needs severe corrections. First, the manuscript is difficult to read mostly due to necessary English corrections and beginning each sentence with a cited author`s name in the Introduction and Discussion.
However, the main concern is the need for a proper repeat of experiments, primarily that they were performed in the field conditions. So, at least two-season experiments are necessary to show.

Comments on the Quality of English LanguageModerate editing of the English language is required.
Some sentences could be clearer to understand. Some sentences are long and should be divided.
I recommend that a native speaker should check and correct this manuscript.
Reviewer 2 Report
Comments and Suggestions for Authors
The research refers to the rice production and significant problems with the premature senescence of leaves and their mechanisms.
Before the publication the work need some corrections. The work should be rewritten taking more care when writing. The introduction should be more concise. There are many repetitions in the text. The varieties should be named according to the nomenclature of cultivated plants.
Line 53: etc?
Lines 58-67: lack of the references. However, this problem is related to the another part of the Introduction text.
Why the Huazhan were named HZ, but Nekken2 hasn’t abbreviation? However, I propose retire the abbreviation. It's not needed and creates confusion.
The aims and novelty of work can be mentioned clearly at the end of the introduction.
The Result also need correction. The Figures 1 and are unreadable. Figure 3 is incomprehensible including title. The text is generally poorly described and should be also rewritten.
I suggest write correctly the results and then rethinking the discussion. The discussion showed the same problems as Introduction – some part without references (?), a few repetitions.
The more conclusions is needed and indications to the further work.
Materials and Methods: the more information about plant material is needed.
Comments on the Quality of English LanguageThe language should be more concise. The parts of the manuscript should be well edited and appear harmonious with each other.